# Early-Onset Colorectal Cancer: Current Insights

**DOI:** 10.3390/cancers15123202

**Published:** 2023-06-15

**Authors:** Fauzia Ullah, Ashwathy Balachandran Pillai, Najiullah Omar, Danai Dima, Seema Harichand

**Affiliations:** 1Department of Translational Hematology and Oncology Research, Cleveland Clinic Foundation, Cleveland, OH 44195, USA; najiullahomar@gmail.com (N.O.); dimad@ccf.org (D.D.); 2Department of General Internal Medicine, The University of Texas MD Anderson Cancer Center, Houston, TX 77030, USA; abalachandran1@mdanderson.org; 3Department of Hematology and Medical Oncology, Taussig Cancer Institute, Cleveland Clinic Foundation, Cleveland, OH 44195, USA; 4Department of Internal Medicine, Mission Cancer + Blood, University of Iowa, Des Moines, IA 50309, USA; sharichand@missioncancer.com

**Keywords:** early-onset colorectal cancer, genetic makeup, modifiable risk factors, Lynch syndrome

## Abstract

**Simple Summary:**

Early-onset colorectal cancer is increasing in incidence in the United States, and it is expected to more than double in the next 10 years. There are many risk factors that lead to the development of early-onset colorectal cancer, including hereditary cancer syndromes, such as Lynch syndrome, obesity, diet, sedentary lifestyle, inflammatory bowel disease and altered microbiome. There is a significant lack of awareness of early-onset colorectal cancer, and the rise in the number of cases indicates that more coordinated efforts are needed to understand and treat early-onset colorectal cancer better, through promoting the benefits of screening, including genetic profiling, and increasing adherence to current screening guidelines. The aim of this review is to discuss the available literature regarding early-onset colorectal cancer to better define the risk factors, histopathology, genetic makeup and management.

**Abstract:**

Over the past decade, the incidence of colorectal cancer has increased in individuals under the age of 50 years. Meanwhile, the incidence has gradually decreased in the older population. As described herein, we reviewed the available literature to summarize the current landscape of early-onset colorectal cancer, including risk factors, clinicopathological presentation, genetic makeup of patients, and management. Currently, early-onset colorectal cancer is treated similarly as late-onset colorectal cancer, yet the available literature shows that early-onset colorectal cancer is more aggressive and different, and this remains a significant unmet need. A detailed understanding of early-onset colorectal cancer is needed to identify risk factors for the increased incidence and tailor treatments accordingly.

## 1. Introduction

Early-onset colorectal cancer (EOCRC) is increasing in incidence, and colorectal cancer (CRC) unfortunately remains the third leading cause of cancer-related deaths in the U.S in both males and females [1]. It is expected to cause around 52,550 deaths in 2023. Approximately 10% of all new diagnoses of CRC are early-onset, and for this reason, the U.S Preventive Services Task Force and the American Cancer Society have decreased the recommended age to initiate screening from 50 years to 45 years [2,3]. Recent studies demonstrated that the incidence of CRC is increasing in adults younger than 50 years by almost 1.4% annually, while there is a gradual decline in adults over 50 years of age by almost 3.1% annually [4]. It is estimated that in the next 10 years, 25% of rectal cancers and 10–12% of CRC will be diagnosed in individuals younger than 50 years of age [5,6]. The incidence of EOCRC is increasing globally [7,8,9,10]. Furthermore, the younger age population tends to have a more advanced stage of disease at diagnosis, aggressive tumor characteristics and more years lost from the impact of the disease when compared to older age groups. Approximately 30% of cases of EOCRC can be attributed to hereditary cancer syndromes, such as Lynch syndrome, familial adenomatous polyposis (FAP), MUTYH-associated polyposis (MAP), and various hamartomatous polyposis conditions. Conversely, about 50% of EOCRC cases are sporadic in onset and are not related to traditional risk factors [6]. Herein, we will discuss the available literature regarding the EOCRC to better define the risk factors, histopathology, genetic makeup and management.

## 2. Non-Modifiable Risk Factors for Early-Onset Colorectal Cancer

The most important risk factor for CRC is family history, and having a first-degree relative with CRC diagnosed under the age of 50 increases the risk of developing CRC by more than 2-4-fold [11]. A significant proportion of CRC cases are sporadic (70%) and familial (25%), and only 5% of cases are due to inherited syndromes [12]. A discussion of EOCRC would not be complete without a review of the well-described literature of hereditary cancer syndromes, which may be divided into polyposis and nonpolyposis syndromes.

### 2.1. Polyposis Syndromes

Patients with familial adenomatous polyposis (FAP) have nearly 100% risk of CRC, and those with attenuated familial adenomatous polyposis (afap) have close to 69% risk of CRC [13,14]. In both syndromes, the APC gene is mutated at chromosome 5q21. In MUTYH-associated polyposis (MAP), the risk of CRC is around 19% by age 50 and increases to 43% by age 60 [15,16,17]. Peutz–Jeghers syndrome confers a risk of around 39% of CRC, with a mutation in the STK11 gene [16,18]. Juvenile Polyposis syndromes with mutations in SMAD4 and BMPR1A genes confer a 10–38% lifetime risk of CRC [13,19]. The exonuclease domain of polymerases, POLE and POLD1 associated adenomatous polyposis confer an unknown risk of CRC and constitutes a minority of familial cases [16,17,20]. Mutations in MSH3 are associated with MSH3-associated polyposis, PTEN gene mutation with hamartoma tumor syndrome and TP53 mutation with Li–Fraumeni syndrome [17,21]. In addition, BRCA1, BRCA2, CHEK2, ATM, and PALB2 gene mutations have all been identified as pathogenic variants in hereditary cancer syndromes, which could impart low to moderate colon cancer risk [17,22].

Furthermore, serrated polyposis syndrome (SPS), as defined by the World Health Organization, is the most prevalent polyposis syndrome. It is characterized by two criteria, (I) the presence of five or more serrated lesions/polyps larger than or equal to 5 mm proximal to the rectum, with at least two lesions larger than or equal to 10 mm; and (II) the occurrence of at least 20 serrated lesions/polyps of any size throughout the large bowel, with at least five of them located proximal to the rectum [16,23,24]. Serrated polyposis syndrome is associated with mutations in the RNF43 gene. The risk of CRC in people with SPS was assessed in a meta-analysis that included 36 studies and 2788 patients. The risk was 14.7% (95% CI, 11.4–18.8%), which was lower than initially reported [25].

### 2.2. Nonpolyposis Syndrome

Hereditary nonpolyposis CRC also known as Lynch syndrome, accounts for 2–4% of all CRCs and the lifetime CRC risk is estimated to be 40–80% [13,14]. It is the most common hereditary CRC syndrome. People with Lynch syndrome have a significantly increased risk of CRC and several other types of cancers. The mean age of onset of CRC is 44–52 years, and it is believed that adenoma–carcinoma transition happens at a much faster rate in Lynch syndrome. This increases the chance that a new neoplasm can appear in 2–3 years after a negative colonoscopy. Stoffel and colleagues evaluated 147 families with lynch syndrome and showed a cumulative risk of CRC of 66% in males and 43% in females [26]. Males with mutations in MLH1 had the highest risk of CRC.

The Amsterdam and the Revised Bethesda guidelines were proposed to identify individuals at risk for Lynch syndrome [16,27,28]. However, limiting patients to the Bethesda criteria only would miss 28% of cases with Lynch Syndrome [29]. Universal screening is recommended for all patients with CRC who have germline mutations in the mismatch repair (MMR) genes to identify patients with Lynch syndrome and refer them accordingly for genetic counseling. Once identified with Lynch syndrome, a colonoscopy is recommended every 1–2 years for surveillance. Furthermore, female patients with Lynch syndrome are advised to have a total hysterectomy and bilateral salpingo-oopherectomy when childbearing is complete [30].

Despite rigorous colonoscopy surveillance leading to increased detection rates, approximately 1.2 million Americans affected by Lynch syndrome are undiagnosed [10,31,32]. It is for this reason that the American Society of Clinical Oncology (ASCO) and the National Comprehensive Cancer Network (NCCN) recommend universal screening for Lynch syndrome in CRC. Even then, there are gaps in achieving improved screening. Part of this is attributed to incomplete family history in the patient medical chart. As part of the ASCO quality oncology practice initiative, Wood et al. conducted a pilot qualitative measures test and found that only about 22% of the patient charts had sufficient family history to identify patients to refer them for screening [33]. Studies have recognized that correctly identifying family history as a risk factor could have potentially prevented 25% of CRC incidence [34].

Additionally, inflammatory bowel disease and cystic fibrosis are known risk factors for EOCRC, hence the reason for existing guidelines to start screening early for this population for CRC [35]. History of radiation therapy in the pelvic and abdominal areas might also be a risk factor for EOCRC. Hadjiliadis et al. recommend early screening starting at age 40 for patients with cystic fibrosis and at 30 years for patients with cystic fibrosis and organ transplant [35]. More frequent surveillance intervals are also recommended at 3–5-year intervals. A retrospective study of patients evaluated at NYU Langone Health reviewed 269 patients with EOCRC, 2802 with late-onset CRC (LOCRC) and 1122 controls [36]. Patients with EOCRC were more likely to have inflammatory bowel disease (3% vs. 0.4% for controls, *p* < 0.01), be male and have a family history of CRC compared with controls (odds ratio (OR) of 1.87, and 8.61), respectively. Furthermore, patients with EOCRC were more likely to be male, black, Asian, have inflammatory bowel disease, or have a family history of CRC; OR of 1.44, 1.73, 2.60, 2.97, and 2.87, respectively. EOCRC was more common in the left colon or rectum (75% vs. 59%, *p* = 0.02) and presented at an advanced stage of tumor growth (77% vs. 62%, *p* = 0.01) than in late-onset disease. The prevalence values of the most common modifiable risk factors (obesity, diabetes, and smoking) of CRC were similar. The results of this study suggest that non-modifiable risk factors, including sex, race, inflammatory bowel disease and family history of CRC, are associated with EOCRC.

## 3. Modifiable Risk Factors for Early-Onset Colorectal Cancer

Modifiable risk factors remain debated, but it is well-studied that a Western diet, smoking, obesity, sedentary lifestyle and the consumption of red and processed meat are risk factors for CRC [10,37,38,39]. These risk factors have been studied primarily in older age groups but have also been identified as risk factors in the younger population. Figure 1 illustrates the most common risk factors for the development of EOCRC.

### 3.1. Obesity and Sedentary Lifestyle

Obesity has been shown to be an independent risk factor for CRC due to the induction of an inflammatory state. There has been an exponential increase in obesity in the U.S in recent years [40,41]. Studies have found an increased risk of CRC in men by 30% and women by 12% with each 5 kg/m^2^ increase in body mass index (BMI) [42]. Obesity causes an increased production of adipocytes and cytokine (tumor necrosis factor and IL-6), which may damage cellular DNA, increase angiogenesis and promote cell proliferation. Furthermore, obesity causes a state of insulin resistance, there is increased insulin and insulin-dependent growth factors [43,44]. Several studies have shown that obesity in adolescence and type 2 diabetes are associated with increased incidence of EOCRC and related mortality. The observation of increased cumulative risk by about 5 years when compared to the general population has been seen in people with diabetes for the risk of CRC [42,45]. Similar reports of increased risk in people with diabetes have been shown in a Swedish cohort study [46].

Investigators have widely studied the benefits of exercise in the primary prevention and management of CRC. A sedentary lifestyle has been shown to have a close association with the increased incidence of CRC. Smoking and alcohol consumption are other known risk factors. Heavy alcohol use conferred a relative risk of almost 1.71 to EOCRC [47,48]. Studies have suggested an association between TV viewing (sedentary time) and the risk of CRC. One of the studies showed that women with highly sedentary and less physically active lifestyles had a 41% increased risk of CRC compared with those more active [38,49]. We also must consider the racial disparities involved in screening for CRC. While we have seen an increase in screening efforts, there still seems to be a gap between Caucasian and non-Caucasian ethnicities [50]. Several factors could contribute to this, including lack of health literacy, access to health care, and poor socio-economic status.

A retrospective study of U.S Veterans ages <50 years analyzed 651 patients with EOCRC and 67,416 controls [51]. Compared with the controls, the median age was 45.3 years, and the majority were older males, current smokers, non-aspirin users and had lower BMIs. Male sex and increasing age were strongly associated with the risk of EOCRC. In a post hoc analyses, the odds of EOCRC were higher if the patient had a weight loss of ≥5 kg within the 5-year period before the colonoscopy (OR 2.23; 95% CI 1.76–2.83). This could be because body weight is measured at study initiation, and weight loss before the diagnosis of CRC is not accounted for. In a subsequent study of 6264 CRC patients and 6866 controls who were recruited from 2003 to 2020 in Germany for CRC screening [52], smoking exposure was significantly associated with both EOCRC (<55 years, 724 cases, 787 controls) and LOCRC (≥55 years, 5540 cases, 6079 controls). The adjusted ORs for EOCRC and LOCRC were as follows: current smoking was 1.57 (*p* < 0.001) and 1.46 (*p* < 0.001), and former smoking was 1.39 (*p* = 0.01) and 1.24 (*p* < 0.001). These findings were similar for cancers of the colon and rectum in patients with early- and late-stage disease. However, more research is needed to understand the factors that cause EOCRC and to identify those at the highest risk.

### 3.2. Western Diet

There is growing evidence that a Western diet remains an important risk factor for CRC worldwide [53]. Western diet has been shown to alter the gut microbiome, causing mucosal inflammation and predisposing individuals to CRC when compared with a plant-based diet [54]. Specifically, eating red meat or processed meat, diets high in saturated fat and low in fiber, and deep-fried foods contain carcinogenic advanced glycation end-products (AGEs) [55]. Red meat/processed meat contain N-nitroso compounds that can harm the colon epithelial lining, causing carcinogenic changes. A systematic review of the dietary influences on early-onset colorectal cancer by Puzzono et al. highlights the effect of gut epithelial disruption deleterious effects on the DNA caused by these meat preservatives [56]. These practices also cause changes to the gut microbiome, which has been shown to play a role in the development of CRC [57]. The dietary inflammatory index has been proposed by researchers to indicate the inflammatory nature of diets, with the Mediterranean diet having the least inflammatory potential and the fewest AGEs [58,59,60,61].

Dietary additives represent another risk factor to consider when considering the recent shift in the increased incidence of EOCRC. Manufacturers are known to increase the number of additives to food to increase its shelf life and attractiveness through the use of food coloring or artificially increase the flavor. Compounds commonly implicated include nitrates and nitrites in processed meat, the consumption of which can cause the occurrence of N-nitroso compounds that are precarcinogenic [62]. Synthetic dyes are added to food to increase its appeal are also considered CRC risk factors as these compounds are differentially metabolized by the gut microbiome and could have the potential to be carcinogenic. Notably, the European Union requires a warning label on foods containing synthetic dyes, whereas countries outside of the European Union, including the U.S, do not. Thus, more rigorous studies are needed in the present context. A lack of dietary fiber is another proposed risk factor for CRC. When fermented by the gut microbiome, dietary fiber produces small-chain fatty acids as an energy source. With a diet low in fiber, the gut microbiome utilizes glycoprotein in the mucus layer, leading to a break in the mucosal layer and potentially exposing the gut epithelial lining to carcinogens [63].

Multiple recent studies have reported an association between CRC and high-fructose corn syrup. The consumption of sugar-sweetened beverages has recently increased in the U.S, which is partly contributing to the obesity epidemic. In the Nurses’ Health Study II from 1991 to 2015, the investigators found that a Western diet was associated with an increased risk of early-onset and high-risk colorectal adenomas, especially in the distal colon and rectum. Furthermore, they found a twofold increase in the risk of EOCRC in women having more than two sugar-sweetened beverages a day. They did not find a similar association with artificially sweetened beverages and fruit juices [55,64]. Of note, this study found that exposure to high-fructose corn syrup increased CRC tumor growth in mice models. This may tie in with the fact that high levels of sweet beverages cause excessive calories and weight gain. It contributes to hyperinsulinemia, obesity, and metabolic syndrome, which cause an increase in circulating inflammatory cytokines, serving as a risk factor for cancer. This finding is of vital importance as young adults are the primary consumers of carbonated beverages with high-fructose corn syrup, which means early exposure could prove detrimental with the increasing incidence. Further studies are needed on dietary causation to determine any potential preventative measures for EOCRC.

### 3.3. Gut-Microbiome

Early-onset CRC has a multifactorial risk profile, including lifestyle and environmental exposures over a long period that can alter the gut microbiome. The gut microbiota is vital in maintaining homeostasis as it plays an important role against pathogens [65]. Research on the association between CRC and the gut microbiome is ongoing, but there is abundant evidence that links the gut microbiome with the development of CRC [6,45,66,67]. Researchers have identified and compared differences in the gut microbiome in people with adenomas versus a control population. Common microbiota in the gut that are thought to be potentially associated with CRC are Fusobacterium nucleatum and Pepto streptococcus anaerobius, which can activate inflammatory pathways and cause altered immune responses. Additionally, certain bacteria, such as Clostridium butyricum and streptococcus salivanes, may have a role in maintaining the normality of the gut flora. Processed and fried foods increase sulfur-metabolizing bacteria in the gut, which, in turn, produces genotoxic products that lead to inflammation and DNA damage, contributing to the risk of CRC [63,68]. Several prospective cohort studies have shown that a Western diet is associated with an excess of sulfur-metabolizing bacteria in feces, which was positively associated with an increased risk of distal CRC [69]. Furthermore, individuals with CRC exhibit a decrease in bacterial diversity compared to the healthy population. Research has indicated that CRC is associated with an abundance of certain bacterial taxa, including Firmicutes, Bacteroidetes, enterotoxigenic Bacteroides fragilis, and oral anaerobes, such as Fusobacterium nucleatum [70,71,72]. The decrease in gut bacteria results in the reduced production of short-chain fatty acids, which play a crucial role in maintaining intestinal immune homeostasis. Additionally, the prolonged use of antibiotics disrupts the gut microbiome, increasing the risk of EOCRC [73,74,75]. Nevertheless, the link between dysbiosis and EOCRC is not clear, and further research is needed to elucidate this association.

## 4. Genetic and Molecular Landscape of Colorectal Cancer

EOCRC is a heterogeneous disease with both hereditary and sporadic components. The pathogenesis of CRC is complex and involves multistep genetic mutations. Several colon carcinogenesis pathways have been proposed, with the three most common pathways being chromosomal instability (CIN), microsatellite instability (MSI) and CpG island methylator phenotype (CIMP) or serrated pathway that is characterized by hypermethylation via the inactivation of tumor suppressor genes [76]. Less than 30% of CRCs are due to a family history of CRC, and only 3–5% are attributed to inherited CRC syndromes [77]. Hereditary colonic polyposis syndromes, such as familial adenomatous polyposis, which result from a germline alteration in the adenomatous polyposis coli (APC) gene and MUTYH-associated polyposis, are caused by biallelic pathogenic variants and predispose individuals to increased risk of CRC, although these are less common than Lynch syndrome [78,79,80,81]. The APC gene dysfunction activates the Wnt pathway, causing the accumulation of beta-catenin, which is involved in the prevention of cell apoptosis. Furthermore, it signals cell adhesion and proliferation. Knudson’s two-hit model mechanism describes this Adenoma Carcinoma Sequence (5 q loss of APC, KRAS mutation, 18 q loss of DCC, 17 p loss of p53) [82]. Patients with EOCRC tend to have higher rates of genetic alterations in the genes predisposing to CRC, including MSH2, MSH6, PTEN and BRCA2 [83]. Approximately 85% of sporadic CRCs are attributed to the CIN pathway, characterized by mutations in the APC, KRAS and TP53 genes. The loss of DNA mismatch repair genes (MMR), which can be germline or sporadic, results in microsatellite instability (MSI) [84]. Lynch syndrome is due to germline mutations in one of the MMR genes (MLH1, MLH3, MSH2, MSH6, PMS2, PMS6 or EPCAM) [84,85]. The serrated polyp or CIMP pathway is associated with BRAF mutation [86]. Figure 2 summarizes the three most common pathogenesis pathways of some of the genomic profiles reported in the literature, and it is important to note that this is not a comprehensive list of all mutations associated with CRC [13,16,76].

The molecular profile of CRC is still debated; however, some evidence suggests that EOCRC tumors are microsatellite-stable (MSS), have a strong familial component and are localized in the left colon when compared with LOCRC [87]. It has been reported that a subset of EOCRCs can exhibit both microsatellite and chromosomal stability (MACS) and is associated with aggressive tumor characteristics, including early metastasis, disease recurrence and poor survival [88]. Compared with EOCRC, LOCRC has a high prevalence of mutations in BRAF V600, NRAS, KRAS and APC genes [89]. Conversely, EOCRC patients are more likely to have mutations in CTNNB1 (encodes beta-catenin), ATM, and hypermethylation of ESR1, GATA5 and WT1 genes [89,90]. Recent reports indicate that EOCRC is more likely to display MSI-H compared with LOCRC, which is strongly associated with poor tumor differentiation [91]. Most MSI-H EOCRC is due to Lynch syndrome, whereas LOCRC is most often associated with sporadic MSI-H from MLH1 hypermethylation and frequent BRAF V600 E mutations [92,93]. A study conducted at the Memorial Sloan Kettering Cancer Center analyzed 759 patients with EOCRC, and after excluding subjects with MSI-H or a predisposition to CRC, either clinical or hereditary, they found no significant differences between patients with EOCRC and LOCRC in terms of genomic profiles or tumor grade [94]. In a similar study of 18,218 CRC cases, next-generation sequencing was performed using a panel comprising 403 cancer-related genes. The results showed that APC, KRAS and BRAF mutations were more prevalent in subjects older than 50 years. Conversely, mutations in TP53 and CTNNB1 were more frequently observed in subjects younger than 40 years of age. However, molecular changes in microsatellite-stable (MSS) cancers were similar across age groups [95].

Moreover, KRAS mutations reportedly occur in up to 50% of cases of CRC in the general population. The presence of KRAS mutations is a strong predictor of a lack of response of CRC to epidermal growth factor receptor (EGFR)-targeted therapy [96]. In the CRYSTAL trial, patients with metastatic CRC expressing EGFR were randomly assigned to receive irinotecan, fluorouracil and leucovorin (FOLFIRI) alone or in combination with cetuximab as a first-line therapy. The results showed a statistically significant improvement in the overall response rate and median progression-free survival among patients with KRAS wild-type tumors who received the cetuximab-containing regimen compared to those with KRAS mutants [97]. RAS proteins are encoded by three different genes, KRAS, NRAS and HRAS, and they are an integral component of cell signaling mechanisms. They are involved in regulating cell proliferation and cell death. These mutations are also predictors for CRC tumor response to EGFR chemotherapy. NRAS mutations G12D and Q61K have been distinctly identified in EOCRC. Researchers have identified NRAS mutations also show resistance to EGFR therapy, making the use of targeted therapy very challenging [98]. PIK3CA mutations are identified in 10–18% of CRC patients. These mutations confer a greater risk of transformation to cancer and predict poor response to treatment [99].

The current guidelines recommend screening all newly diagnosed metastatic CRC patients for MSI and MMR status due to the robust response to immune checkpoint inhibitors as a first-line therapy [100]. The Dana-Farber Cancer Institute conducted a prospective study involving 1058 patients who underwent germline testing of twenty-five genes. The results revealed that 9.9% of the patients had one or more pathogenic variants, and among those, 31.4% were identified as having Lynch syndrome [77]. This was undertaken without preselection for age, family history or MSI status. Approximately 31.8% of the sample size had EOCRC, and of those, 14.0% had at least one germline variant, with 44.7% having Lynch syndrome. The results of the study suggest the presence of pathogenic germline variants in approximately 17% of patients with EOCRC, and half of these variants were in MMR genes associated with Lynch syndrome. It is for this reason that genetic risk counseling and testing by next-generation sequencing is highly recommended by the NCCN and Collaborative Group of the Americas on Inherited Gastrointestinal Cancer to include the MMR genes that are associated with increased risk of cancer in carriers (APC, MUTYH, BMPR1A, SMAD4, PTEN and STK11) [78,101]. These six genes cause various forms of polyposis.

Despite an increase in the incidence of EOCRC in recent years, there is a lag in understanding the molecular pathogenesis of CRC. We have not been able to identify any unique features that could explain the shift from LOCRC to EOCRC, and this remains an unmet need. Establishing a good predictive model will allow clinicians to risk stratify patients according to their molecular profile, and in order to do this, more retrospective and prospective randomized studies are needed to elucidate the genetic profile of EOCRC.

## 5. Clinical and Histopathologic Characteristics of CRC

EOCRC is different from LOCRC in terms of the clinical presentation and the tumor characteristics. The clinical features of EOCRC include decreased appetite, weight loss, abdominal pain, rectal bleeding, anemia and changes in bowel habits [102,103,104]. Moreover, EOCRC is predominantly in the left colon, with the more common location of the tumor being in the recto–sigmoid region [105]. A multicenter retrospective study found that more than 61% of EOCRC patients presented with metastatic disease at the time of diagnosis compared with 44% of LOCRC patients [102,106]. This could be due to aggressive tumor characteristics, including a genomic profile that predisposes to accelerated carcinogenesis in EOCRC [107]. A subset of CRC tumors (<40%) are mucinous or have signet ring cell features. EOCRCs also tend to be larger (>5 cm at diagnosis) and have more involved lymph nodes, and are more likely to have perineural involvement at diagnosis than LOCRCs [102,108,109]. A multidisciplinary international group (DIRECt) put together evidence-based recommendations to guide clinicians caring for patients with EOCRC [110].

There could be an element of both patient and provider factors in a delayed diagnosis in the younger population. When younger age group presents with rectal bleed, without a family history or when it is not properly documented/elicited, CRC may not be the first differential considered. This could contribute to a delay in referral for colonoscopy. There could also be a delay in the patient seeking care due to lack of awareness [111].

## 6. Screening

The current guidelines recommend screening starting at the age of 45 years in the general population. More frequent screening is recommended for people having second-degree relatives with CRC or at least one first-degree relative with CRC before the age of 50 years to start screening by 40 years [110]. A study of EOCRC from the Ohio CRC Prevention Initiative assessed that with early screening, at least 16% of the patients could have been diagnosed earlier [34]. The DIRECt guidelines recommend assessing the CRC risk, workup of symptoms and colonoscopy be performed within 30 days of presentation [110,112]. There are several screening options for CRC. Providers should encourage their patients to follow screening guidelines, as the early detection of CRC can save lives and reduce morbidity. The most common screening options include colonoscopy, flexible-sigmoidoscopy, CT colonography, fecal immunochemical test (FIT), stool DNA test and fecal occult blood test (FOBT), which can be combined with sigmoidoscopy and FIT [113,114]. Colonoscopy remains the gold standard for CRC screening. A mutual and individualized decision-making approach is strongly encouraged. Figure 3 depicts the most common CRC screening options.

## 7. Treatment

Compared to LOCRC, EOCRC is associated with aggressive tumor characteristics, more advanced stage, systemic therapy use and is more likely to be managed aggressively [112]. There are no specific evidence-based treatment protocols for EOCRC, and therapeutic strategies are being implemented based on the patient’s age, tumor stage and comorbidities. The current National Comprehensive Cancer Network (NCCN) guidelines recommend the same treatment regimens for EOCRC and LOCRC in both the palliative and curative settings [115]. In addition, the current guidelines from DIRECt recommend similar systemic treatment for EOCRC and LOCRC [110]. Surgery is the mainstay treatment for CRC, and EOCRC patients tend to receive radiotherapy at all stages and aggressive adjuvant chemotherapy, even for early-stage disease, when compared with LOCRC, with only marginal benefit [109,116]. However, long-term survival is still debated. A multicenter study reported a better prognosis for EOCRC compared to LOCRC [117]. Conversely, a single-center retrospective study showed worse recurrence, PFS and cancer-specific survival for EOCRC compared to LOCRC [118]. A multicenter randomized trial [119] and a retrospective study from the Memorial Sloan Kettering Cancer Center [94] showed no significant difference in survival between patients with metastatic EOCRC and those with metastatic LOCRC. The comparative results concerning long-term survival are unclear, and large multicenter retrospective and randomized prospective trials are needed to better understand the outcome and treatment regimens for patients with EOCRC. Table 1 shows ongoing prospective studies of EOCRC.

## 8. Care Considerations in Survivors

The increasing incidence of EOCRC has led to unique survivorship concerns in this patient population, including sexual dysfunction, fertility, and body image [120]. The type of surgery performed may have a significant impact with a probable worse outcome regarding body image in patients undergoing abdominoperineal excision and ostomy placement in comparison to the alternative procedures [121]. Furthermore, men can experience erectile dysfunction. Care teams should tailor care to individual patients and appropriately direct them to the necessary resources to address these concerns.

Fertility in patients with EOCRC is another concern. For example, treatment with 5-FU chemotherapy can decrease sperm count and may cause amenorrhea [122]. The effects of drugs, such as oxaliplatin, irinotecan and anti-EGFR and vascular endothelial growth factor (VEGF) therapy on fertility remains unknown. Pregnancy is another consideration with careful planning of treatment needed, particularly during the first trimester. Carefully weighing the pros and cons of 5-FU and oxaliplatin has been used in the second and third trimesters [123]. It is important to evaluate the patient as a whole instead of focusing only on their disease.

## 9. Conclusions

The incidence of EOCRC is increasing in the U.S, prompting a decrease in the age of screening. The incidence of CRC is projected to steadily increase and more than double by 2030 [5]. EOCRC has distinct clinical characteristics, pathogenesis and aggressive tumor behavior compared with LOCRC. With the more aggressive tumor presentation of EOCRC, more productive years are lost to treatment, and there are greater economic impacts experienced than in those with LOCRC, and the overall impact at a personal level for each patient is more. If lifestyle changes and dietary habits contribute to the increasing incidence of EOCRC, then educational efforts should be directed toward creating public awareness because research has shown that modifying risk factors could reduce mortality by 12% over a 20-year period [124]. The rise in the number of cases of EOCRC indicates that more coordinated efforts are needed to understand and treat EOCRC better. Efforts to promote the benefits of and adherence to screening, including genetic profiling, should be considered for all patients to guide treatment strategies and counseling. Further research is needed to understand the underlying cause and mechanisms of EOCRC.

## Figures and Tables

**Figure 1 cancers-15-03202-f001:**
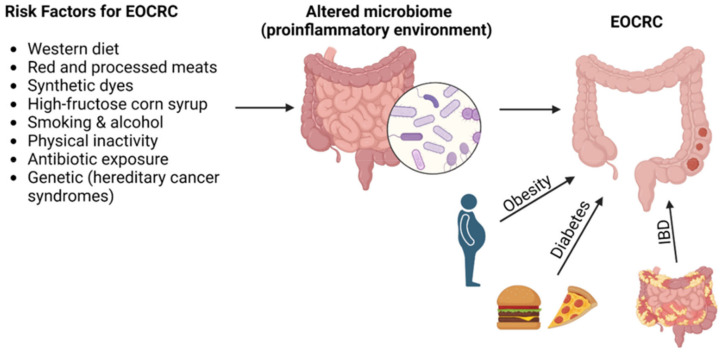
Risk factors for the development of early-onset colorectal cancer. Abbreviations: EOCRC, early-onset colorectal cancer; IBD, inflammatory bowel disease.

**Figure 2 cancers-15-03202-f002:**
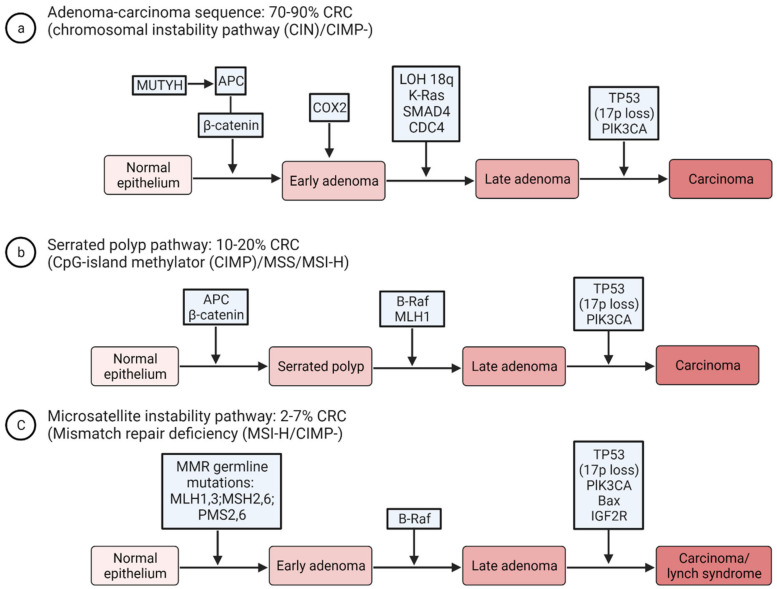
Molecular pathogenesis and classification of CRC. (**a**) key mutations that are required for progression along the adenoma–carcinoma sequence in the chromosomal instability (CIN) pathway. Progression in this pathway with mutations in K-ras, etc., and TP53 leads to carcinoma. (**b**) The DNA mismatch repair (MMR) gene MLH1 can be inactivated either by a mutation or by promoter hypermethylation, which typically occurs in the context of the CpG island methylator phenotype (CIMP). B-Raf mutations and MLH1 hypermethylation are associated with serrated polyps pathway. (**c**) Key mutations in the microsatellite instability high (MSI-H) pathway. Abbreviations: APC, adenomatous polyposis coli; COX2, cyclooxygenase 2; LOH, loss of heterozygosity; MSS, microsatellite stable.

**Figure 3 cancers-15-03202-f003:**
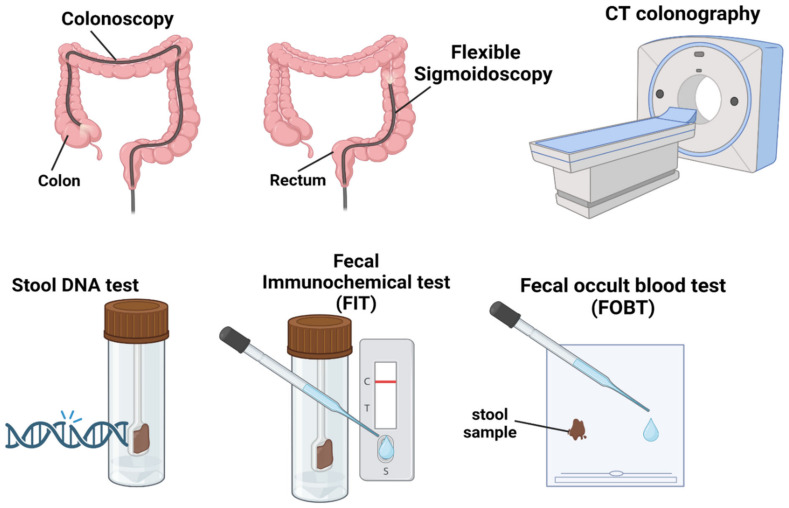
Colorectal cancer screening options.

**Table 1 cancers-15-03202-t001:** Ongoing prospective studies of early-onset colorectal cancer.

NCT Number	Study Title	Intervention	Status
NCT04715074	A community-based intervention to increase EOCRC awareness	Behavior: Interviews	Not yet Recruiting
NCT04812912	Changes in reproductive and sexual health in people with EOCRC	Hormone biomarker analysis and QoL questionnaires	Recruiting
NCT05732623	Exogenous and endogenous risk factors for EOCRC	Semi Quantitative food frequency questionnaire (SQFFQ)	Recruiting
NCT05184751	Incidence and risk factors of EOCRC	Colonoscopy	Completed
NCT02664389	Targeted next-generation sequencing panel for identification of germline mutations in EOCRC with sporadic or hereditary	Genetic analysis	Completed
NCT01057953	Oligogenic determination of colorectal cancer	Blood drawn	Completed
NCT00044967	Genetic study of young patients with colorectal cancer	Microsatellite instability analysis	Completed
NCT03214939	Autologous antigen-activated dendritic cells in the treatment of patients with colorectal cancer	Immunotherapy based on dendritic cells	Unknown

Abbreviations: EOCRC, early-onset colorectal cancer.

## Data Availability

Not applicable.

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
