# Peer review of "Early-Onset Colorectal Cancer: Current Insights"

_cancers, 2023, doi:10.3390/cancers15123202_

Round 1

Reviewer 1 Report

The research object is relevant in today's oncology. Colon cancer is an increasing disease. The work is topical, covering many of the issues being addressed today on this topic. It complements the published publications on this topic, describes in more detail the possible risks of colon cancer. I do not have any fundamental suggestions for improving the methodology for the authors. The conclusions of the work correspond to the questions raised.

Author Response

Dear Editor,

Thank you for the opportunity to submit a revised draft of my manuscript titled "Early-Onset Colorectal Cancer: Current Insights”. We appreciate the time and effort that you and the reviewers have dedicated to providing your valuable feedback on this manuscript. 

Reviewer 1 had no comments. 

Reviewer 2 Report

Dear editors,

 The review by Fauzia Ullah and colleagues aims to describe current insights on early-onset colorectal cancer. However, I must note that there are significant issues with the manuscript that need to be addressed before publication. Specifically, there are several reviews on this topic present in the literature, including in this journal. Thus, to be different and new, the review should analyze new data that has appeared in the literature. The literature revision is incomplete and not updated, which is the most significant limitation of the study. Furthermore, there are several inaccuracies and missing references throughout the manuscript. Therefore, I recommend that the authors carefully revise the manuscript before it can be accepted for publication, and I am willing to review it again once the revisions have been made.

Specifically:

1) Only two papers are cited from 2022 and three from 2023. The literature revision is incomplete and not updated. 

2) Page. 6 lines 49-50: “Interestingly, the age of onset of CRC screening colonoscopy 49 has not changed in Europe (60 years)” This is incorrect. In Europe, CRC screening focuses on asymptomatic individuals more than 50 years of age.

3)    Pag 5, DIET. Once again, the topic is lacking several important references and leads to incorrect conclusions. I must suggest to better study and cite the review on this topic that appeared on Cancers. 2021 Nov 25;13(23):5933. doi: 10.3390/cancers13235933. PMID: 34885046

4) Pag 5 Lifestyle: Despite previous reports linking physical activity (or lack thereof) to eoCRC, an approximately equal weight of evidence to the contrary exists at present. Please comment.

5) Pag 6: Genetic and molecular landscape of colorectal cancer. This topic seems to be written by non-experts in the field of onco-genetics. The first paragraph report very superficial, not useful details containing some inaccuracies (eg: “Expression of mismatch repair genes is downregulated in hereditary cancer syndromes like Lynch syndromes). 

6) Pag 9-10-11: Characteristics of EOCRC versus LOCRC; Treatment and Care considerations in survivors. These topics must be discussed based on the clinical guidelines recently published: Clin Gastroenterol Hepatol 2023 Mar;21(3):581-603.e33. doi: 10.1016/j.cgh.2022.12.006. Epub 2022 Dec 20

Author Response

Dear Editor,

Thank you for the opportunity to submit a revised draft of my manuscript titled "Early-Onset Colorectal Cancer: Current Insights”. We appreciate the time and effort that you and the reviewers have dedicated to providing your valuable feedback on this manuscript. Below is a summary of the incorporated changes reflecting the suggestions provided by the reviewers. We have highlighted the changes within the manuscript.

Here is a point-by-point response to the reviewer’s comments and concerns.   

Comments from Reviewer 2

Comment 1: Only two papers are cited from 2022 and three from 2023. The literature revision is incomplete and not updated. 

Response: We agree, the review was completely revised and now it includes more than 10 references from the last few years. The references were added throughout the revised manuscript.

Comment 2: Page. 6 lines 49-50: “Interestingly, the age of onset of CRC screening colonoscopy 49 has not changed in Europe (60 years)” This is incorrect. In Europe, CRC screening focuses on asymptomatic individuals more than 50 years of age.

Response: Thank you for pointing this out. This was removed

Comment 3: Pag 5, DIET. Once again, the topic is lacking several important references and leads to incorrect conclusions. I must suggest to better study and cite the review on this topic that appeared on Cancers. 2021 Nov 25;13(23):5933. doi: 10.3390/cancers13235933. PMID: 34885046

Response: The review was completely revised to include important references. This specific reference can be found on page 5, under the section “western diet” 5th sentence of the first paragraph.

Comment 4: Pag 5 Lifestyle: Despite previous reports linking physical activity (or lack thereof) to eoCRC, an approximately equal weight of evidence to the contrary exists at present. Please comment.

Response: This was removed to reduce confusion

Comment 5: Pag 6: Genetic and molecular landscape of colorectal cancer. This topic seems to be written by non-experts in the field of onco-genetics. The first paragraph report very superficial, not useful details containing some inaccuracies (eg: “Expression of mismatch repair genes is downregulated in hereditary cancer syndromes like Lynch syndromes). 

Response: We agree this section was poorly written. It was revised and checked against multiple references. The inaccuracies were removed.

Comment 6: Pag 9-10-11: Characteristics of EOCRC versus LOCRC; Treatment and Care considerations in survivors. These topics must be discussed based on the clinical guidelines recently published: Clin Gastroenterol Hepatol 2023 Mar;21(3):581-603.e33. doi: 10.1016/j.cgh.2022.12.006. Epub 2022 Dec 20

Response: This reference was added and can be found in the following sections, “clinical and histopathologic characteristics of CRC” and “screening”.

Sincerely,

Dr Fauzia Ullah

Reviewer 3 Report

Early-Onset Colorectal Cancer: Current Insights

A brief summary

In this review manuscript, the authors have investigated the environmental and genetic factors involved in the early incidence of colorectal cancer.

Specific comments

1. One of the keywords is polyposis syndrome, which is not mentioned in the abstract or the simplified abstract. It is better to write a little about this term in the abstract.

2. In the Figure 1(on the left), it is better to write “Genetic” instead Genetics.

3. Bring the comparison of environmental and genetic factors involved in late and early colorectal occurrence in a separate title.

4. Explain more and in detail about the treatment differences between late and early colorectal cancer

Author Response

Dear Editor,

Thank you for the opportunity to submit a revised draft of my manuscript titled "Early-Onset Colorectal Cancer: Current Insights”. We appreciate the time and effort that you and the reviewers have dedicated to providing your valuable feedback on this manuscript. Below is a summary of the incorporated changes reflecting the suggestions provided by the reviewers. We have highlighted the changes within the manuscript.

Here is a point-by-point response to the reviewer’s comments and concerns.   

Comments from Reviewer 3

Comment 1: One of the keywords is polyposis syndrome, which is not mentioned in the abstract or the simplified abstract. It is better to write a little about this term in the abstract.

Response: The section was updated with additional terms.

Comment 2: In the Figure 1(on the left), it is better to write “Genetic” instead Genetics.

Response: This was corrected.

Comment 3: Bring the comparison of environmental and genetic factors involved in late and early colorectal occurrence in a separate title.

Response: The review was completely revised, and this distinction is made in the sections – “Genetic and molecular landscape of colorectal cancer”, 2nd paragraph and “Clinical and Histopathologic Characteristics of CRC”.

Comment 4: Explain more and in detail about the treatment differences between late and early colorectal cancer.

Response: Please refer to the revised treatment section as this was updated.  

Sincerely,

Dr Fauzia Ullah 

Round 2

Reviewer 2 Report

None